# Effect of the COVID-19 Pandemic on Rates and Epidemiology of *Clostridioides difficile* Infection in One VA Hospital

**DOI:** 10.3390/antibiotics12071159

**Published:** 2023-07-07

**Authors:** Lorinda M. Wright, Andrew M. Skinner, Adam Cheknis, Conor McBurney, Ling Ge, Susan M. Pacheco, David Leehey, Dale N. Gerding, Stuart Johnson

**Affiliations:** 1Edward Hines Jr. VA Hospital, 5000 S. 5th Ave., Hines, IL 60141, USA; andrew.skinner@va.gov (A.M.S.); adam.cheknis@va.gov (A.C.); conor.mcburney@va.gov (C.M.); ling.ge@va.gov (L.G.); susan.pacheco2@va.gov (S.M.P.); david.leehey@va.gov (D.L.); dale.gerding2@va.gov (D.N.G.); stuart.johnson2@va.gov (S.J.); 2Chicago Stritch School of Medicine, Loyola University, 2160 S. First Ave., Maywood, IL 60153, USA

**Keywords:** *Clostridioides difficile*, COVID-19, SARS-CoV-2

## Abstract

The COVID-19 pandemic was associated with increases in some healthcare-associated infections. We investigated the impact of the pandemic on the rates and molecular epidemiology of *Clostridioides difficile* infection (CDI) within one VA hospital. We anticipated that the potential widespread use of antibiotics for pneumonia during the pandemic might increase CDI rates given that antibiotics are a major risk for CDI. Hospital data on patients with CDI and recurrent CDI (rCDI) were reviewed both prior to the COVID-19 pandemic (2015 to 2019) and during the pandemic (2020–2021). Restriction endonuclease analysis (REA) strain typing was performed on CD isolates recovered from stool samples collected from October 2019 to March 2022. CDI case numbers declined by 43.2% in 2020 to 2021 compared to the annual mean over the previous 5 years. The stool test positivity rate was also lower during the COVID-19 pandemic (14.3% vs. 17.2%; *p* = 0.013). Inpatient hospitalization rates declined, and rates of CDI among inpatients were reduced by 34.2% from 2020 to 2021. The mean monthly cases of rCDI also declined significantly after 2020 [3.38 (95% CI: 2.89–3.87) vs. 1.92 (95% CI: 1.27–2.56); *p* = <0.01]. Prior to the pandemic, REA group Y was the most prevalent CD strain among the major REA groups (27.3%). During the first wave of the pandemic, from 8 March 2020, to 30 June 2020, there was an increase in the relative incidence of REA group BI (26.7% vs. 9.1%); After adjusting for CDI risk factors, a multivariable logistic regression model revealed that the odds of developing an REA group BI CDI increased during the first pandemic wave (OR 6.41, 95% CI: 1.03–39.91) compared to the pre-pandemic period. In conclusion, the incidence of CDI and rCDI decreased significantly during the COVID-19 pandemic. In contrast, REA BI (Ribotype 027), a virulent, previously epidemic CD strain frequently associated with hospital transmission and outbreaks, reappeared as a prevalent strain during the first wave of the pandemic, but subsequently disappeared, and overall CDI rates declined.

## 1. Introduction

*Clostridioides difficile* (CD) is a spore-forming, anaerobic, gram-positive bacteria. CD infection (CDI), the most common cause of healthcare-associated infection (HCA) [1], is an ongoing major global public health concern and typically occurs after antibiotic use disrupts the normal gut microbiota. Despite recent downward trends in HCA-CDI, CDI is considered an urgent threat, causing 235,700 cases in hospitalized patients, 16,200 deaths, and an estimated USD 1 billion in healthcare costs annually [2,3]. The incidence of CDI increased during the 2000s, concurrently with the spread of a novel hypervirulent strain designated PCR ribotype (RT) 027, North American Pulse Field 1 (NAP1), or restriction endonuclease analysis (REA) type BI. Despite effective treatment of the initial infection, 15–35% of CDI cases spontaneously recur after the cessation of initial therapy [4,5].

The Edward Hines Jr. Veterans Affairs Hospital (HVAH) participates in the VA Cooperative Studies Program (CSP) trial, CSP #596: “Optimal Treatment of Recurrent *C. difficile* Infection (OpTION).” As part of this study, pre-screening data are collected on all HVAH patients with positive CD stool test results and/or who are treated for CDI. Patient records are tracked to identify cases of recurrent CDI (rCDI) within 90 days after successful treatment.

The rapid global spread of SARS-CoV-2, the virus that causes coronavirus disease 2019 (COVID-19), resulted in the declaration of a national emergency in the United States and nationwide shutdowns on 13 March 2020. These pandemic shutdowns markedly altered hospitalizations, even for non-COVID-19 patients. In order to study the interaction of COVID-19 and CDI, we examined rates of CDI, rCDI, and CD stool testing in 2020 and 2021 at our hospital in comparison to prior years. We also performed REA strain typing on stool samples collected prior to and during the COVID-19 pandemic.

## 2. Results

### 2.1. CD Stool Testing

Prior to the COVID-19 pandemic, from 1 January 2015 to 31 December 2019, a mean of 15.12 CDI cases/month were diagnosed and/or treated at HVAH (Table 1). During this time, a mean of 15.45 submitted stool tests were designated as positive for CD by nucleic acid amplification testing (NAAT) monthly. Annually, a mean of 140.0 unique patients had NAAT+ stool test results in 2015–2019 (*n* = 700), and a mean of 136.0 unique patients were treated for CDI during this time (*n* = 680). Beginning in early 2020 and continuing throughout 2021, there was a significant 35.3% decline in monthly NAAT+ stool tests to a mean of 10.00 (*p* < 0.01). This reduction in monthly NAAT+ stool tests remained consistent in both pandemic years, with a mean of 9.33 (*p* < 0.01) monthly NAAT+ tests in 2020 and a mean of 10.67 (*p* < 0.01) monthly NAAT+ tests in 2021.

The number of monthly CDI diagnoses also declined significantly in 2020–2021 to a mean of 8.58 (*p* < 0.01), representing a 43.2% fall in monthly CDI cases compared to the previous 5 years. Monthly CDI diagnoses were also consistently lower in both 2020 (8.17 CDI diagnoses/month (*p* < 0.01) and 2021 (9.00 CDI diagnoses/month (*p* < 0.01). The mean annual numbers of unique patients with NAAT+ stool testing (92.0; *n* = 184) and unique CDI patients (75.0; *n* = 150) were also lower in 2020–2021 compared to the previous 5 years.

The mean monthly number of rCDI cases was also significantly lower in 2020–2021 (1.92 vs. 3.38; *p* < 0.01), representing a 43.4% decrease in monthly CDI recurrence. In the 5 years preceding the pandemic, a total of 203 recurrences were documented among 907 CDI cases (22.4%) within 3 months. The rate of CDI recurrence was similar overall in 2020–2021 (46 recurrences out of 206 CDI cases; 22.3%). However, a lower percentage of CDI cases recurred in the first year of the pandemic (15 recurrences out of 98 CDI cases; 15.3%), while the rate of CDI recurrence increased in 2021 (31 recurrences out of 108 CDI cases; 28.7%. The mean monthly number of CDI recurrences was significantly lower in 2020 (1.25; *p* < 0.01), but not in 2021 (2.58; *p* = 0.18).

A comparison of total CD positivity rate (fraction of all stool tests that were CD+ by NAAT) revealed a significant decline from a mean of 17.2% positivity in 2018 and 2019 (393 NAAT+ of 2287 tests run) to 14.3% positivity during 2020–2021 (240 NAAT+ of 1681 tests run; *p* = 0.01). The CD positivity rate was lower in both 2020 (112 NAAT+ of 789 tests run; 14.2%; *p* = 0.05) and 2021 (128 NAAT+ of 892 tests run; 14.4%; *p* = 0.05). Data for 2015–2017 were unavailable for comparison. Throughout the analysis period, there was no change in the small percentage of cases that were treated for CDI without confirmatory stool testing.

Prior to the implementation of reflex enzyme immunosorbent assay (EIA) toxin stool testing in September of 2019, CD colonization was diagnosed rarely, and most NAAT+ patients were treated for CDI. After the implementation of stool toxin testing, which was performed using a NAAT+ stool test, CD colonization was more frequently diagnosed when toxin testing was negative coupled with a documented lack of ongoing diarrhea and/or presence of an identifiable alternative source of diarrhea. From 2015–2019, a mean of 0.33 cases/month were diagnosed with CD colonization and were, thus, not treated for CDI (Table 1). After implementation of the two-step reflex into the EIA toxin testing algorithm in September 2019, monthly CD colonization diagnoses increased by more than fourfold in 2020–2021 to 1.42 cases/month (*p* < 0.01). The number of monthly CD colonization diagnoses were significantly higher in both 2020 (1.17; *p* < 0.01) and in 2021 (1.67; *p* < 0.01). Throughout our study period, CDI diagnoses were made for cases that required medical treatment for CD infection, regardless of toxin positivity status. While the increased CD colonization diagnoses in 2020–2021 reduced the proportion of all NAAT+ cases that were treated for CDI, it did not account for the overall reduction in NAAT+ stool testing, CDI, and rCDI during this time period. Our hospital documented totals of only 112 NAAT+ stool tests in 2020 and 128 NAAT+ stool tests in 2021 compared to a mean of 185.4 NAAT+ stool tests/year over the previous 5 years (Figure 1a). The daily case count of SARS-CoV-2 + patients at HVAH from 17 March 2020 to 31 March 2022 is shown for comparison in Figure 1b. The daily case count of SARS-CoV-2+ patients at HVAH during the first pandemic wave, from 17 March 2020 to 30 June 2020, is shown in Figure 1b (inset).

The CDI treatment guidelines at HVAH changed during the study period, according to recommendations published by the Infectious Diseases Society of America (IDSA) in 2018. During 2015–2019, treatment of patients with metronidazole was common (36% of patients), while oral vancomycin was used in the majority of patients (57%) and fidaxomicin was rarely used (1.3% of patients). Changes in the CDI treatment guidelines resulted in fewer patients being treated with metronidazole in 2020–2021 (8.3%), while the majority of patients (68%) continued to receive oral vancomycin. Oral fidaxomicin use increased slightly (3.9%).

The overall annual HVAH patient volume decreased during the pandemic years. There was a 15.6% decline in unique inpatient admissions in 2020–2021 compared to the previous 5 years (5009/year vs. 5933/year; *p* < 0.01), while the outpatient volume remained unchanged (57,597 vs. 57,102 unique outpatients/year, Figure 2a)**.** However, there was a significant decline (34.2%) in the frequency of CDI diagnoses among inpatients in 2020–2021 (0.78% vs. 1.19% of inpatient admissions, *p* < 0.01), as well as a 41.2% decline in CDI diagnoses among unique outpatients per year (0.058% vs. 0.098%, *p* < 0.01), in 2020–2021 compared to the previous 5 years (Figure 2b). The number of NAAT+ stool tests declined similarly in 2020–2021 compared to the previous 5 years, among both inpatients (0.94% vs. 1.21% of inpatients, *p* < 0.01) and outpatients (0.062% vs. 0.10% of outpatients, *p* < 0.01).

The majority of CDI diagnoses were made among hospital inpatients, and there were no changes in the relative frequency of inpatient diagnosis (69%) compared to outpatient diagnosis (31%) during the period of our analysis. The relative rates of healthcare-onset (HO)-CDI (40%), community-onset (CO)-CDI (47%), and healthcare facility-associated (CO-HCFA) CDI (13%) diagnoses also remained constant from 2015 to 2021.

Overall antibiotic usage at HVAH declined among outpatients during the COVID-19 pandemic, but was unchanged among inpatients. Prior to the pandemic (1, April 2019–31 March 2020) the mean number of outpatient antibiotic prescription fills per 1000 unique patients was 2.08 (95% CI: 1.95–2.20) compared to 1.61 (95% CI: 1.45–1.77; *p* < 0.01) during the pandemic (1 April 2020–30 September 2021). In contrast, inpatient antibiotic usage did not significantly change during this same period: 329.36 antibiotic days/1000 inpatient days (95% CI: 273.56–385.16) prior to the pandemic vs. 327.43 (95% CI: 299.77–355.08; *p* = 0.93) during the pandemic.

### 2.2. REA Typing and Patient Characteristics

We compared mean patient age, white blood cell count (WBC), serum creatinine, serum albumin, and body temperature on the day of NAAT+ CD stool testing for all patients at HVAH from 1 October 2019 to 31 March 2022 (Table 2). During the first wave of the COVID-19 pandemic (Figure 1b), there was a significant increase in mean body temperature compared to the period prior to the COVID-19 lockdown (37.3C vs. 36.8C; *p* < 0.01). There were no significant changes in mean patient age, serum creatinine, or serum albumin during the study period. The results were similar when only patients with active CDI were considered and patients with CD colonization were excluded from the analysis.

We performed REA typing of stool samples collected between 1 October 2019 and 31 March 2022. Prior to 8 March 2020 REA group Y was the most prevalent CD strain, accounting for 27.3% of isolates (*n* = 12; Table 2). During the initial wave of the COVID-19 pandemic (8 March 2020–30 June 2020), there was an increase in the prevalence of REA group BI (26.7% vs. 9.1%; *p* = 0.17). Although the small number of BI infections did not contribute to an overall increase in CDI recurrence or deaths within 90 days, there was evidence of increased severity. Of four CDI cases in which group BI was recovered during this initial pandemic wave, two developed subsequent CDI recurrence and one died within three months. There was also a concomitant decline in prevalence in REA group Y (6.7% vs. 27.3%). The incidence of group BI subsequently declined (26.7% vs. 11.0%) after the initial COVID-19 wave (1 July 2020–31 March 2022; *p* = 0.09). After adjusting for risk factors for CDI, the odds of developing an REA group BI CDI was found to be higher during the first wave of the COVID-19 pandemic when compared to the pre-pandemic time period (aOR: 6.41, 95% CI: 1.03–39.91; Table 3). There was a subsequent decrease in the odds of developing an REA group BI infection after the first wave of the COVID-19 pandemic (aOR: 0.20, 95% CI: 0.04–0.99).

## 3. Discussion

An abrupt and significant decline in both CDI and rCDI diagnoses was observed at HVAH beginning in April 2020. Others have similarly reported declines in CDI diagnosis during the first years of the COVID-19 pandemic [6,7], including within the entire VA healthcare system [8]. In contrast, some reports have found lower levels of CO-CDI, but not HO-CDI [9], while others have reported either no significant change [10] or modestly increased CDI rates in 2020 and 2021 [11]. Some have suggested that the rates of CD testing, rather than actual infections, declined [12], or that patients were more likely to delay seeking care early in the pandemic [13]. While our study was limited to a single large VA hospital, our analysis captured both CDI testing and treatment and found no increase in empiric treatment of CDI. Reduced CDI rates at HVAH are, therefore, unlikely to be related solely to lower testing volume or delays in seeking treatment. The relevance of the decline in CDI should be considered in light of hospitalizations for COVID-19 and the high rate of antibiotic use in these patients. A review of 1007 abstracts found that the majority (58–95%) of COVID-19 inpatients received empiric antimicrobial treatment to prevent ventilator-associated pneumonia and other secondary infections, despite low rates (8%) of bacterial or fungal coinfections [14].

Co-infection with CDI and SARS-CoV-2 has occurred in a minority of patients with significant co-morbidities, despite the high rates of antibiotic use in COVID-19 patients [14,15]. We found this to be the case at HVAH, with only six co-infected patients in 2020 and 2021, all of whom had histories of significant co-morbidities, including cancer, chronic kidney disease, diabetes, hypertension, chronic obstructive lung disease, and hepatitis. Improved isolation measures in COVID-19 wards and antimicrobial stewardship programs potentially helped to prevent outbreaks of CDI [16]. Conversely, several other hospital-acquired infections were significantly associated with COVID-19 hospitalizations [7].

Infection control measures to prevent the spread of COVID-19, including limiting visitors, increased hand hygiene, use of personal protective equipment, application of universal precautions, and isolation of COVID-19 patients on restricted wards may have helped to reduce the spread of CDI in healthcare settings [17]. Likewise, community masking policies, business and school closures, prohibitions on indoor gatherings, and increased hand hygiene and cleaning may have reduced the spread of CDI and respiratory viral infections [18,19,20].

Beginning on 31 March 2020, all non-essential healthcare visits at HVAH were canceled or conducted via Telehealth. Nationwide, there were striking declines in virtually all non-COVID-19-related healthcare encounters, including visits to the emergency department [21], outpatient hospital visits [22], surgeries [23], and even hospitalizations for acute myocardial infarctions [24], which continued throughout 2020. HVAH hospital census data show a significant reduction in inpatient admissions that may have partially accounted for the decline in CDI among inpatients, although there was no such decrease in outpatient volume to account for the decline in CDI among outpatients. Furthermore, we saw no indication of a shift toward CO-CDI with reduced healthcare usage and inpatient volume at HVAH. It is also unlikely that changes in the treatment of CDI at HVAH accounted for the decreased rates of CDI and rCDI in particular, since fidaxomicin use was limited.

There were nationwide increases in antibiotic prescriptions in hospitals with higher numbers of COVID-19 cases, but a sharp decline in nationwide outpatient antibiotic prescriptions, which was coincident with the decrease in overall healthcare usage in 2020 [25]. Nationwide outpatient antibiotic prescriptions returned nearly to pre-pandemic levels by June 2021. We also noted a decrease in outpatient antibiotic prescriptions at the HVAH during the COVID-19 pandemic, but no change in the rate of inpatient antibiotic usage. In contrast to the nationwide data, the rate of inpatient antibiotic use did not increase at the HVAH. Further information on the rates of antibiotics used for the treatment of pneumonia or among patients with and without COVID-19 were not available. The dramatic declines in NAAT+ stool tests, CDI cases, and CDI recurrence seen at HVAH in 2020–2021 remain to be fully understood.

The addition of reflex to EIA toxin testing at HVAH in September 2019 resulted in significantly more patients being diagnosed with CD colonization and, therefore, not being treated for CDI. Others have noted that inappropriate diagnosis of CDI has previously erroneously inflated CDI case numbers [26], potentially resulting in over-treatment of patients without active infections. Our analysis included a distinction between CDI and CD colonization, and reveals a significant decline in all NAAT+ stool tests as well as CDI at HVAH in 2020 and 2021.

We found a significant increase in the prevalence of the hypervirulent CD strain BI (NAP1/RT 027) during the first four months of the pandemic at our hospital. This is in contrast to recent trends showing declining prevalence of this strain through June 2020 [27]. While our data were obtained from a limited sample set, we note that the declining trends in BI shown by Gentry et al. were not apparent in 2020 in the West North Central region of the US, where our hospital is located.

BI is associated with more severe HO-CDI, and with a greater likelihood of CDI recurrence and death [28,29]. Despite the increased prevalence of this strain, we found no change in the likelihood of CDI recurrence from cases that occurred during this time, likely due to the small number of total cases. The increase in BI during the initial pandemic wave was accompanied by a corresponding decrease in REA group Y. Vendrik et al. conducted surveillance of CDI in nine sentinel Dutch hospitals and performed PCR ribotyping on recovered CD isolates. They found an increase in RT 020 and a decrease in RT 014 infections during the first year of the pandemic in the Netherlands that could not be explained by the spread of specific RT 020 clones [13]. Both RT 020 and RT 014 correlate with REA Group Y strains [30].

Beyond a modest increase in mean body temperature, we found no evidence that CDI patients at HVAH were sicker during the pandemic. This finding is unsurprising, since HVAH CDI patients were almost exclusively a separate population from COVID-19 patients, and we found no evidence of greater CDI disease severity, more frequent recurrence, or that patients were delaying seeking care for CDI.

The limitations of our study are that data were obtained from a single large VA hospital, and our findings may not reflect other institutions. Also, 97% of our patients were male. Furthermore, the change in CD stool testing paradigm to include a reflex to toxin testing in September of 2019 had a significant impact on the number of patients who were not treated for CDI despite NAAT+ stool testing. While we have attempted to account for this change by presenting our data on overall NAAT+ stool testing, as well as patients diagnosed with and treated for CDI, this change presents as a confounding factor in our study and suggests that, given the increase in the number of patients diagnosed as “colonized” with CD after September of 2019, CDI may have been over-diagnosed at our hospital prior to this time. Additionally, since this was a retrospective study, we were not able to control for all possible variables that may have introduced confounding factors into this study.

Herein, we have shown that the lockdowns resulting from the COVID-19 pandemic were associated with a significant and ongoing decline in CD stool testing, CDI, and CDI recurrence at one VA hospital. During the first wave of the pandemic, there was a brief increase in the prevalence of the hypervirulent hospital-associated CD strain type BI/RT 027 and a compensatory decrease in the proportion of REA Group Y (RT014/020) strains. No subsequent outbreaks were identified, and the overall CDI rates continued to decline. It is likely that the decline in CDI rates at our hospital, as well as nationwide, was due to multiple factors. Further research is needed in order to clarify whether COVID-19 infection control measures, antibiotic usage patterns, or characteristics of hospitalized patients during the pandemic played the most important role in the decreased CDI rates which we have observed.

## 4. Materials and Methods

### 4.1. Data Collection

We performed a retrospective review of all laboratory stool testing for CD and a review of the electronic medical records for CDI treatment at HVAH from 1 January 2015 to 31 December 2021. Most CDI cases (1105 of 1162; 95.1%) were confirmed with NAAT stool testing at HVAH (Xpert CD, Cepheid) or an outside hospital with documentation in the medical records. The remainder were treated empirically as CDI; no stool testing was performed, but upon chart review, these patients had consistent symptoms and responded to specific antibiotic treatment for CDI. Guidelines for the treatment of CDI at HVAH were created according to the most updated IDSA/SHEA treatment guidelines in at the time of diagnosis [31,32]. Prior to 1 September 2019, laboratory diagnosis of CDI utilized NAAT testing only. Based on the 2017 IDSA/SHEA guidelines for CDI testing, we introduced a 2-step algorithm, as there was no institutional agreement to limit stool CD testing for patients taking laxatives or for those who did not have >3 unformed bowel movements within 24 h [31]. Beginning 1 September 2019, a NAAT+ test was reflexed to an EIA toxin test for CD (Cdiff quick check complete, Alere/TechLab). When stool testing was positive by NAAT and negative by CD EIA toxin testing, results were reported with the following clarifiers: “Possible disease or colonization. Interpretation requires clinical judgement.” When stool testing was positive by both NAAT and toxin EIA, the alert read: “Toxigenic *C. difficile* detected. Repeat testing should not be performed.” Repeated stool testing within 7 days was not performed. In cases where stool testing was positive according to NAAT (with or without a positive toxin test) and the medical record documented that treatment for CDI was unnecessary, the case was considered as CD colonization. HO-CDI, CO-CDI, and CO-HCFA-CDI were defined according to standard definitions [33]. We defined rCDI as a second CDI diagnosis ≤90 days after successful initial treatment. A census of unique HVAH inpatients/outpatients was obtained from a search of Ambulatory Care Reporting Project records by Patient Administrative Services. Outpatient antibiotic use at HVAH was obtained from the PBM Power BI Outpatient Antibiotic Use Dashboard on the VHA Antimicrobial Stewardship Task Force SharePoint site.

### 4.2. REA Typing

The NAAT+ stool samples collected during routine testing for CD at HVAH were frozen for subsequent testing, then thawed, inoculated on taurocholate-cefoxitin-cycloserine-fructose agar plates (TCCFA), and incubated for 48–72 h in an anaerobic chamber [34]. Distinct colonies with a typical CD morphology were subcultured onto a BBL anaerobic blood agar and incubated anaerobically for 48–72 h. The CD isolates were frozen at −80 °C prior to subsequent analysis. REA typing was performed on CD isolates as previously described [35]. Briefly, total cellular DNA was subjected to *Hind*III digestion, and DNA fragments were separated by electrophoresis on a 0.7% agarose gel. The resulting restriction patterns were compared with patterns from previously characterized strains. Patterns showing a 90% similarity index were placed in the same REA group. The correlations of REA strain types with PCR Ribotype (RT) designations were shown in parentheses. [30,36].

### 4.3. Statistical Analysis

The χ2 and Fisher’s exact tests were used to compare the prevalence of CDI cases and REA types within groups between time periods. Student’s T-test was used to compare parametric variables, and the Wilcoxon signed-rank test was used to compare non-parametric variables across two dependent variables. If more than two dependent variables were compared, one-way ANOVA was utilized for parametric variables and Kruskal–Wallis for non-parametric variables. A value of *p* < 0.05 was considered significant.

A series of multivariable logistic regression models were constructed to evaluate the relationships between the time periods defined as: (1) pre-COVID (1 October 2019–7 March 2020); (2) initial COVID wave (8 March 2020–30 June 2020); and (3) subsequent COVID waves (1 July 2020–31 March 2022). CDI was attributed to REA group BI, Y, and DH strains individually. The dates defining the initial COVID wave were chosen in order to capture all CDI cases that occurred during the week of the declaration of a national emergency on 13 March 2020, and until the end of the first spike in daily COVID-19 cases at our hospital. The variables included in the model included antimicrobial exposures within 6 months prior to CDI diagnosis which previously demonstrated a high risk for CDI: (1) β-lactam antibiotics (cephalosporins, carbapenems, aminopenicillins, β-lactamase antibiotics); (2) macrolides; and (3) fluroquinolones. All other antibiotic exposures were grouped together as “other antibiotics”, excluding metronidazole and IV vancomycin. Metronidazole exposure was included as a separate covariate. Additionally, the model included age, immunocompromised status, and proton pump exposure. Immunocompromised status was defined as a history of hematologic malignancy, active chemotherapy, or immunomodulating medications. Patients with two or more separate clinical episodes of NAAT+ testing were considered to have undergone separate “encounters.” Statistical analyses were conducted using SAS statistical software v9.4 (SAS Institute, Cary, North Carolina). The results were reported as adjusted odds ratios (aORs) with 95% confidence intervals (CIs).

## 5. Conclusions

Herein, we have shown that the COVID-19 pandemic lockdown was associated with a significant and ongoing decline in *Clostridioides difficile* positive stool testing and CD infection rates at one VA hospital. We also found a temporary increase in the prevalence of the hypervirulent CD strain type BI (NAP1/RT 027) in the early stages of the pandemic, but no subsequent outbreaks, and the overall CDI rate decreased.

## Figures and Tables

**Figure 1 antibiotics-12-01159-f001:**
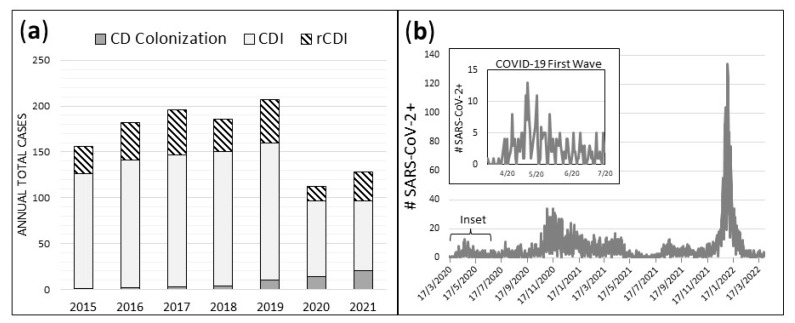
Total annual cases of CD colonization, CDI, and rCDI (**a**), and daily case count of SARS-CoV-2+ patients at HVAH from 17 March 2020 to 31 March 2022 (**b**).

**Figure 2 antibiotics-12-01159-f002:**
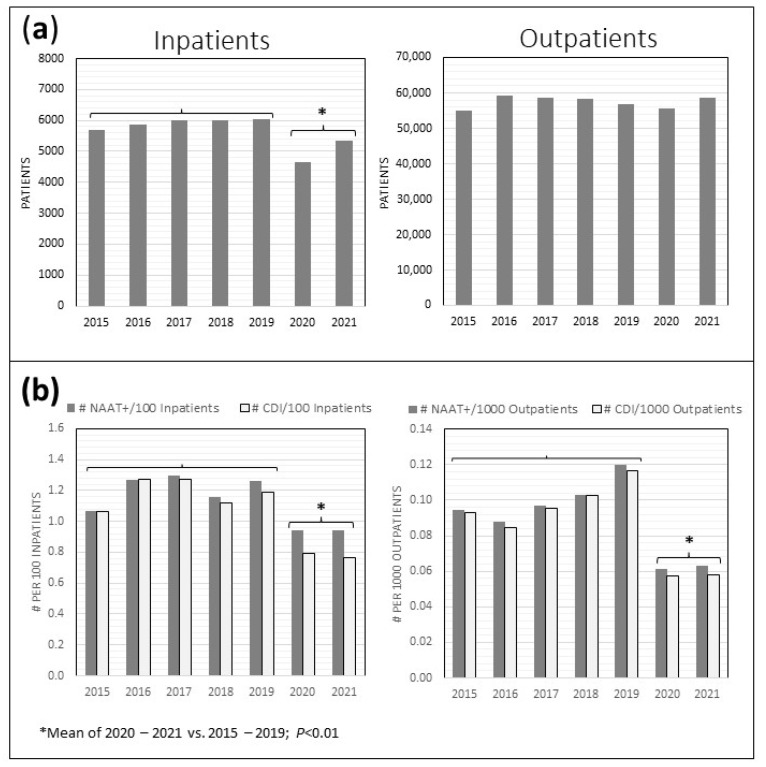
HVAH inpatient and outpatient census (**a**) and rate of CDI diagnosis and NAAT+ stool tests among inpatients and outpatients (**b**). Inpatients include those in the acute care wards and the contiguous extended care facility.

**Table 1 antibiotics-12-01159-t001:** Total CD stool testing at HVAH from 2015 to 2021.

	Pre-Pandemic (2015–2019)	First Pandemic Year (2020)	*p*-Value *	Second Pandemic Year (2021)	*p*-Value ^†^	Pandemic Years (2020–2021)	*p*-Value ^‡^
Mean Monthly NAAT ^1^+ Tests (95% CI; *n*)	15.45 (14.53–16.37; *n* = 927)	9.33 (7.79–10.87; *n* = 112)	<0.01	10.67 (9.06–12.28; *n* = 128)	<0.01	10.00 (8.94–11.06; *n* = 240)	<0.01
Mean Monthly Total CDI ^2^ Cases (95% CI; *n*)	15.12 (14.20–16.03; *n* = 907)	8.17 (6.90–9.43; *n* = 98)	<0.01	9.00 (7.54–10.46; *n* = 108)	<0.01	8.58 (7.68–9.49; *n* = 206)	<0.01
Mean Monthly rCDI ^3^ Cases (95% CI; *n*)	3.38 (2.89–3.87; *n* = 203)	1.25 (0.53–1.97; *n* = 15)	<0.01	2.58 (1.55–3.61; *n* = 31)	0.18	1.92 (1.27–2.56; *n* = 46)	<0.01
No. Total NAAT+ CD ^4^ Stool Tests in 2018–2021 (Total Stool Tests Run; %NAAT+)	393 (2287; 17.2%)	112 (789; 14.2%)	0.05	128 (892; 14.4%)	0.05	240 (1681; 14.3%)	0.01
Mean Monthly CD Colonization (95% CI; *n*)	0.33 (0.15–0.52; *n* = 20)	1.17 (0.51–1.82; *n* = 14)	<0.01	1.67 (0.98–2.35; *n* = 20)	<0.01	1.42 (0.97–1.86; *n* = 34)	<0.01

^1^ NAAT: Nucleic acid amplification test; ^2^ CDI: CD infection; ^3^ rCDI: recurrent CDI; ^4^ CD: *Clostridioides difficile.* * Comparing 2020 vs. 2015–2019; **^†^** Comparing 2021 vs. 2015–2019; **^‡^** Comparing 2020–2021 vs. 2015–2019.

**Table 2 antibiotics-12-01159-t002:** CD strain REA typing, CDI outcome, and case characteristics.

	All Encounters	Pre-Pandemic (1 January 2019 – 7 March 2020)	Initial Pandemic Wave (8 March 2020 – 30 June 2020)	Post-Initial Wave (1 July 2020 – 31 March 2022	*p*-Value
Case Characteristics	(*n* = 327)	(*n* = 82)	(*n* = 32)	(*n* = 213)	
Median Age (IQR ^1^)	73 (67–78)	73.0 (67–82)	73.0 (68–82)	73.0 (67–77)	0.96 *
No. Male (%)	316 (96.6)	77 (93.9)	31 (96.9)	208 (97.7)	0.28 ^†^
No. Immunocompromised (%)	63 (19.3)	9 (11.0)	7 (21.9)	47 (22.2)	0.09 ^†^
No. PPI ^2^ (%)	192 (58.9)	45 (55.0)	19 (59.4)	128 (60.4)	0.69 ^†^
Median No. Days Prior Antibiotics Exposure (IQR)	14 (3–29)	15 (3–35)	19 (6–36)	14 (2–27)	0.24 *
Mean Temperature, C (95% CI)	36.8 (36.7–36.8)	36.8 (36.6–37.0)	37.3 (36.9–37.7)	36.7 (36.6–36.7)	<0.01 ^‡^
Mean WBC ^3^ (95% CI)	10.5 (9.8–11.2)	11.3 (9.7–12.8)	8.7 (7.3–10.0)	10.5 (9.6–11.4)	0.17 ^‡^
Mean Creatinine (95% CI)	2.0 (1.8–2.3)	1.7 (1.4–2.0)	1.4 (1.1–1.8)	2.3 (1.9–2.6)	0.05 ^‡^
Mean Albumin (95% CI)	2.6 (2.5–2.7)	2.4 (2.2–2.6)	2.7 (2.5–2.9)	2.6 (2.5–2.8)	0.14 ^‡^
REA ^4^ Strain Typing	(*n* = 159)	(*n* = 44)	(*n* = 15)	(*n* = 100)	
No. REA Group Y (RT ^5^ 014/020) (%)	32 (20.1)	12 (27.3)	1 (6.7)	19 (19.0)	0.21 ^†^
No. REA Group BI (RT 027) (%)	19 (12.0)	4 (9.1)	4 (26.7)	11 (11.0)	0.17 ^†^
No. REA Group DH (RT 106) (%)	20 (12.6)	3 (6.8)	2 (13.3)	15 (15.0)	0.39 ^†^
Other REA Groups (%)	88 (55.4)	25 (56.8)	8 (53.3)	55 (55.0)	0.97 ^†^

* Kruskal–Wallis test comparing differences across three time periods; **^†^** chi-square analysis comparing differences across three time periods; **^‡^** one-way ANOVA analysis comparing differences across three time periods. ^1^ IQR: interquartile range; ^2^ PPI: proton pump inhibitors; ^3^ WBC: white blood cell count; ^4^ REA: restriction endonuclease analysis; ^5^ RT: PCR ribotype.

**Table 3 antibiotics-12-01159-t003:** Adjusted odds ratio for key REA groups between time periods.

First Pandemic Wave Compared to Pre-Pandemic Period *	
	Adjusted Odds Ratio (95% CI ^1^)
REA ^2^ Group BI (RT 027)	6.41 (1.03–39.91)
REA Group DH (RT 106)	2.87 (0.35–23.27)
REA Group Y (RT 014/020)	0.13 (0.01–1.26)
Post First Pandemic Wave Compared to First Pandemic Wave ^†^	
REA Group BI (RT 027)	0.20 (0.04–0.99)
REA Group DH (RT 106)	0.35 (0.04–2.82)
REA Group Y (RT 014/020)	3.56 (0.41–31.43)

* Pre-pandemic Period: 1 October 2019–7 March 2020; first pandemic wave: 8 March 2020–30 June 2020; ^†^ post-first pandemic wave: 1 July 2020–31 March 2022. ^1^ CI: confidence interval; ^2^ REA: restriction endonuclease analysis.

## Data Availability

The data presented in this study are available upon request from the corresponding author.

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
