# Peer review of "Effect of the COVID-19 Pandemic on Rates and Epidemiology of Clostridioides difficile Infection in One VA Hospital"

_antibiotics, 2023, doi:10.3390/antibiotics12071159_

Round 1

Reviewer 1 Report

I would like to thank the authors for the review article opportunity.

Comment 1:

The author presented the number of mean cases per month compared pre-COVID-19 and COVID-19 phases (2020-2021)

It was difficult to statistically compare because it might be affected by COVID-19 and decrease hospitalization.

I may be more confident if the author presented in:

-         proportion of  PCR positive C.difficile per admission patients

-         PCR positive per diarrhea illness patients

-         PCR positive in who negative C. difficile toxin

-         PCR positive in those who GDH positive

-         CDI is diagnosed per all admission patients.

-         CDI colonization per all admission patients

It needed to revise, including Figure 1a

Comment 2:

I encourage the authors to declare local PCR CD testing guidelines and provide a rationale. The number of symptomatic c vs. asymptomatic CDI should be noted.

The number of NAATs testing was reduced by more than 60%. Please more discussed.

Comment 3:

The change strain to NAP1/B1 during early COVID-19 was compatible with the outbreak definition. Please provide more data and discussion about the result of the outbreak investigation process, including how it affected the results and how the infection control program during an outbreak.

Comment 4 ;

Please discuss local guideline treatment for CDI and the factors contributing to rCDI, such as duration of treatment, immunosuppressed, severity, outbreak in the hospital, etc.

Comment 5:

Line 101: CD colonization was more frequently diagnosed when toxin testing was negative, coupled with a documented lack of ongoing diarrhea and the presence of an identifiable alternative source of diarrhea.

This definition is invalid and needs to declare and revised.

The authors should cite the reference of the definition according to IDSA guidelines.

Comment 6

Line 110: Throughout our study period, CDI diagnosis was defined as a case that required medical 110 treatment for CD infection regardless of toxin positivity status.

Does It mean many patients are overdiagnosed with CDI?

If yes, please include the discussion on limitations.

Comment 6:

Line 184: The relevance of the decline in CDI should be considered in light of hospitalizations for COVID-19 and the high rate of antibiotic use in these patients. A review of 1007 abstracts found that most (58-95%) of COVID-19 inpatients received empiric antimicrobial treatment to prevent ventilator-associated pneumonia and other secondary infections despite low rates (8%) of bacterial or fungal coinfections.

Please state clearly, why there is a decline in CDI in a high rate of antibiotic use.

Moreover, the previous antibiotic use should be revised into case characteristics in Table 2

Comment 7
We found this to be the case at HVAH with only 6 co-infected patients in 2020 and 2021, all with histories of significant co-morbidities, including cancer, chronic kidney disease, diabetes, hypertension, chronic obstructive lung disease, or hepatitis. Improved isolation measures in COVID-19 wards and antimicrobial stewardship programs potentially helped prevent outbreaks of CDI.

It was possibly true. However, this presented the outbreak of the B1 strain that contrasts with the author's notes.

The author also discussed the rate of stool CD PCR that might be decreased during the SARS-COV-2 outbreak.

Comment 8: Please provide the data on Limitations.

Comment 9:

Line 317: the conclusion: we have shown that the pandemic lockdowns were associated with a dramatic 317 and ongoing decline in Clostridioides difficile positive stool testing and CD infection rates 318 at one VA hospital. We also found a temporary increase in the prevalence of the hyper-319 virulent CD strain type BI (NAP1/RT 027).

The conclusion needed to revise. This study showed a decrease in the proportion of  Group Y during the COVID-19 phase. The increase in the proportion of B1 might be small CDi outbreaks.

Author Response

The authors would like to thank the reviewers for their thoughtful comments. We would like to offer the following in response to the reviewers’ questions:

Reviewer 1

Comment 1: The author presented the number of mean cases per month compared pre-COVID-19 and COVID-19 phases (2020-2021). It was difficult to statistically compare because it might be affected by COVID-19 and decrease hospitalization….

Response: We agree with the reviewer that several potential factors may confound the observed decrease in CDI cases during the pandemic. In addition to case rates, we also looked at stool positivity rates (# of CD positive stools/# stools tested), hospitalization rates, and outpatient encounters. At this reviewer’s suggestion, we also analyzed the proportion of PCR-positives per inpatient admission and per outpatient visit and included this additional data as a third panel in Figure 2. 

Comment 2: I encourage the authors to declare local PCR CD testing guidelines and provide a rationale. The number of symptomatic c vs. asymptomatic CDI should be noted. The number of NAATs testing was reduced by more than 60%. Please more discussed

Response: We have now included information in the methods section regarding CD testing guidelines. Throughout the study period, clinicians submitted stool specimens based on suspicion for CDI. Formed stools were rejected by the clinical laboratory and repeat testing was not allowed for at least 7 days afterwards. After the switch to 2-step testing the results for PCR positive, toxin A/B negative tests were reported as “C. diff detected by PCR, toxin negative by EIA. Possible disease or colonization. Interpretation requires clinical judgement.” PCR positive, toxin A/B positive tests were reported as “Toxigenic C. difficile detected. Repeat testing should not be performed.” The large decrease in the number of NAAT stool tests in 2020 is one of the major observational findings of our study, and we believe this finding indicates an overall decrease in CDI coincident with the COVID-19 pandemic.

Comment 3: The change strain to NAP1/B1 during early COVID-19 was compatible with the outbreak definition. Please provide more data and discussion about the result of the outbreak investigation process, including how it affected the results and how the infection control program during an outbreak.

Response: We agree that this was by definition an outbreak. The REA typing was done well after the clinical events and there was no specific infection control measure put in place for this outbreak which resolved spontaneously. Review of medical charts for the 4 patients with a BI infection during the initial Covid-19 wave did not reveal any commonalities in their residence or treatment locations prior to the CDI diagnosis at HVAH that may have suggested a common source of the infections.

Comment 4: Please discuss local guideline treatment for CDI and the factors contributing to rCDI, such as duration of treatment, immunosuppressed, severity, outbreak in the hospital, etc.

Response: We have now included in the methods section a brief summary of local guidelines for treatment of CDI, as well as a paragraph summarizing patient CDI treatments in the results. Local HVAH treatment guidelines were according to the IDSA/SHEA treatment guidelines in place at the time [1, 2]. Most patients receive oral vancomycin. Oral metronidazole was used much less frequently after 2018, and fidaxomicin was reserved for those deemed at increased risk for recurrent CDI. Data on immunosuppressed patients is presented in Table 2.

Comment 5: CD colonization was more frequently diagnosed when toxin testing was negative, coupled with a documented lack of ongoing diarrhea and the presence of an identifiable alternative source of diarrhea. This definition is invalid and needs to declare and revised. The authors should cite the reference of the definition according to IDSA guidelines

Response: We agree with the reviewer that there is no universally agreed upon or guideline-recommend definition of ‘CD colonization’. We used an operational definition of CD colonization as stated in the methods section where the medical record documented that treatment was unnecessary and not administered. The 2017 IDSA/SHEA guidelines for CDI discuss the epidemiology of CDI colonization and infection at length as well as the recommendations for testing to limit unnecessary treatment of patients that are simply colonized. We have now included in the methods section the following explanation/rationale for 2-step testing that was introduced at our hospital: Based on the 2017 IDSA/SHEA guidelines for CDI testing, we introduced the 2-step algorithm as there was no institutional agreement to limit stool CD testing for patients taking laxatives or with those who did not have >3 unformed BMs in 24 hr [1].

Comment 6: Line 110: Throughout our study period, CDI diagnosis was defined as a case that required medical treatment for CD infection regardless of toxin positivity status. Does It mean many patients are overdiagnosed with CDI? If yes, please include the discussion on limitations.

Response: Undoubtedly ‘over diagnosis’ of CDI occurred at our hospital, even after implementation of the 2-step testing algorithm. We attempted to control for this by looking at PCR positive results before and after the 2-step testing. We have now included a separate paragraph in the discussion devoted to limitations.

Comment 6B: Line 184: The relevance of the decline in CDI should be considered in light of hospitalizations for COVID-19 and the high rate of antibiotic use in these patients. A review of 1007 abstracts found that most (58-95%) of COVID-19 inpatients received empiric antimicrobial treatment to prevent ventilator-associated pneumonia and other secondary infections despite low rates (8%) of bacterial or fungal coinfections. Please state clearly, why there is a decline in CDI in a high rate of antibiotic use. Moreover, the previous antibiotic use should be revised into case characteristics in Table 2.

Response: We now have added some additional data to regarding overall inpatient and outpatient antibiotic prescriptions at our hospital.  We speculate in the discussion about potential correlations between antibiotic use and CDI at our hospital. There was a clear decline in overall outpatient antibiotic use beginning near the start of the pandemic, but virtually no change in inpatient antibiotic use during this time. It is difficult to draw specific conclusions regarding the relationship between local inpatient and outpatient antibiotic use and subsequent CDI without more information on specific antibiotics and indications (e.g., pneumonia) for antibiotic use which was not available to us.  We have also revised Table 2 to include prior antibiotic exposure with patient characteristics. There was no significant change in prior antibiotics exposure throughout our study period in patients who had NAAT+ stool testing at our hospital.

Comment 7: We found this to be the case at HVAH with only 6 co-infected patients in 2020 and 2021, all with histories of significant co-morbidities, including cancer, chronic kidney disease, diabetes, hypertension, chronic obstructive lung disease, or hepatitis. Improved isolation measures in COVID-19 wards and antimicrobial stewardship programs potentially helped prevent outbreaks of CDI. It was possibly true. However, this presented the outbreak of the B1 strain that contrasts with the author's notes. The author also discussed the rate of stool CD PCR that might be decreased during the SARS-COV-2 outbreak.

Response: The outbreak of CDI due to BI was small and during the very first wave of the COVID-19 pandemic. Again, this outbreak was only recognized well after the clinical events when the strain typing was performed. It is possible that infection control practices evolved during the pandemic and limited subsequent outbreaks. Overall stool testing declined during the pandemic, the rate of positivity among tests performed declined, and there was no indication of a surge in CDI, or increase in CDI severity after the COVID-19 lockdowns had been lifted.

Comment 8: Please provide the data on Limitations

Response: We have now included a separate paragraph in the discussion listing the limitations of our study.

Comment 9: Line 317: the conclusion: we have shown that the pandemic lockdowns were associated with a dramatic and ongoing decline in Clostridioides difficile positive stool testing and CD infection rates at one VA hospital. We also found a temporary increase in the prevalence of the hyper- virulent CD strain type BI (NAP1/RT 027). The conclusion needed to revise. This study showed a decrease in the proportion of Group Y during the COVID-19 phase. The increase in the proportion of B1 might be small CDi outbreaks

Response: We have revised our conclusions to mention the compensatory decrease in proportion of REA Group Y (RT014/020) strains as well as the increase in Group BI (NAP1/RT 027) strains, emphasize that this occurred early during the first wave of the pandemic and that as the pandemic evolved, no further outbreaks were identified and that the rate of CDI overall decreased.

  1. McDonald LC, Gerding DN, Johnson S, Bakken JS, Carroll KC, Coffin SE et al. Clinical Practice Guidelines for Clostridium difficile Infection in Adults and Children: 2017 Update by the Infectious Diseases Society of America (IDSA) and Society for Healthcare Epidemiology of America (SHEA). Clin Infect Dis 2018; 66: 987-994. PMID: 29462280.
  2. Johnson S, Lavergne V, Skinner AM, Gonzales-Luna AJ, Garey KW, Kelly CP et al. Clinical Practice Guideline by the Infectious Diseases Society of America (IDSA) and Society for Healthcare Epidemiology of America (SHEA): 2021 Focused Update Guidelines on Management of Clostridioides difficile Infection in Adults. Clin Infect Dis 2021; 73: 755-757. PMID: 34492699.

Reviewer 2 Report

The manuscript “Effect of the COVID-19 Pandemic on Rates and Epidemiology of 2

Clostridioides difficile Infection in One VA Hospital” by Lorinda M. Wright and colleagues examines rates of CDI, rCDI, and CD stool testing in 2020 and 2021 at their hospital in comparison to prior years by NATT assay and also performed REA strain typing on stool samples collected prior to and during the COVID-19 pandemic.

They found that the lockdowns resulting from the COVID-19 pandemic were associated with a dramatic and ongoing decline in CD stool testing, CDI and CDI recurrence at one VA hospital. 

The incidence of CDI and rCDI decreased significantly during the initial waves of the COVID-19 pandemic, although the potential widespread use of antibiotics for pneumonia during the pandemic might increase CDI rates given that antibiotics are a major risk for CDI.

During the first wave of the pandemic, there was an increase in the prevalence of hypervirulent hospital-associated CD strain type BI/RT 027. 

Line 148-149: “mean body temperature … 99.1F vs. 98.2F…” please report T in Celsius

The authors report the data along the text including the CI, these data are already in the table and in my opinion ruin the reading flow without giving a real support to the narrative.

Since the increase of antibiotic treatments is a key point of the story, authors should include some data regarding this.

Author Response

The authors would like to thank the reviewers for their thoughtful comments. We would like to offer the following in response to the reviewers’ questions:

Comment 1: “Please report T in Celsius.”

Response: Thank you for this suggestion. This change has been made in the table.

Comment 2: “The authors report the data along the text including the CI, these data are already in the table and in my opinion ruin the reading flow without giving a real support to the narrative.”

Response: We have removed some of the confidence intervals from the text in order to improve the language flow.

Comment 3: “Since the increase of antibiotic treatments is a key point of the story, authors should include some data regarding this.”

Response: We agree with this comment. We have added an additional table showing the volume of antibiotics prescribing at our hospital from 2019 – 2021.

Round 2

Reviewer 1 Report

See attachment

Author Response

The authors would like to thank reviewer 1 for their continued effort to improve the presentation of our work. We would like to offer the following in responses:

Comment 1:  The discussion should focus on whether changes in the CDI guidelines contributed to the reduction in recurrent CDI (rCDI) rates.

Response: It is doubtful that this change affected rates, particularly since fidaxomicin use was limited. We now comment on this in the discussion.

Comment 2:  From Figure 2, It should be better if the author showed (a), (b) and (c) in the same picture

Response: Figure 2a is raw numbers, Figures 2b & 2c show rates. We feel it would be inappropriate to merge all 3. We have combined (b) and (c) into a single panel.

Comment 3: I am satisfied with the response however; it should be clear because the standard of hospital care should define an outbreak. I encourage the author to revise, “During the COVID-19 outbreak, the clarification of the CDI outbreak proved challenging due to the diversion of resources for COVID-19. Despite an increase in the NAP1/RT 027 strain, the CDI outbreak spontaneously disappeared, coinciding with an overall decrease in CDI rates. This favorable outcome may be attributed to the implementation of the COVID-19 infection prevention and control program.” instead of “no further outbreaks was identified and that the rate of CDI overall decreased.”

 Response: We disagree. The CDI outbreak was only recognized after the fact when typing of CD strains was performed (not because of resource diversion). In addition, we have no proof that implementation of the COVID-19 infection prevention and control program affected the CDI rates. We have elected to keep our conclusion as stated.

Comment 4: The study outcome should be discussed in relation to the increasing rate of antibiotic usage despite a decrease in the number of inpatients, while the rate of antibiotic use remained unchanged.

Response: In our discussion (lines 230 to 239) we mention nationwide trends in antibiotic use and compared that to the findings in our hospital. As we stated, the overall rate of inpatient antibiotic use was unchanged at our hospital despite the decrease in inpatient census. We have clarified our discussion as follows:

There were nationwide increases in antibiotic prescriptions in hospitals with higher numbers of COVID-19 cases, but a sharp decline in nationwide outpatient antibiotic prescriptions coincident with the decrease in overall healthcare usage in 2020 [25]. Nationwide outpatient antibiotic prescriptions returned to near pre-pandemic levels by June 2021. We also noted a decrease in outpatient antibiotic prescriptions at the HVAH during the COVID-19 pandemic, but no change in the rate of inpatient antibiotic usage. In contrast to the nationwide data, the rate of inpatient antibiotic use did not increase at the HVAH. Further information on rates of antibiotics used for treatment of pneumonia or rates of antibiotics among patients with and without COVID-19 were not available. The dramatic declines in NAAT+ stool tests, CDI cases, and CDI recurrence seen at HVAH in 2020 – 2021 remain to be fully understood.

Comment 5: Line 189:

There was a sudden, dramatic decline in CDI and rCDI diagnosis at HVAH beginning in April 2020.

It should be revised to “An abrupt and significant decline in both CDI and rCDI diagnoses was observed at HVAH, beginning in April 2020.”

Response: We have made the suggested wording change.

Comment 6: The authors should include a brief discussion regarding the importance of "hand washing with soap" during the COVID-19 pandemic, as it played a significant role in preventing CDI. Additionally, it is essential to describe the pattern of isolation implemented at HVAH, as it could be a key feature in controlling CDI during the COVID-19 outbreak. Moreover, the probably discussed less patient touching in HCPs also should be discussed.

I encourage the author to provide the lesson-learned to reduce the CDI rate during the COVID-19 pandemic.

Response: We agree with the reviewer that hand washing with soap during the COVID­-19 pandemic could have played a role in preventing CDI as well as the pattern of isolation. However, we have no proof that these measures played a role in decreasing CDI rates. We have discussed these potential reasons in the discussion (lines 213-219). In our opinion, multiple factors likely played a role and we have included the following sentence at the end of the discussion (line 285): No subsequent outbreaks were identified, and the overall CDI rates continued to decline….. It is likely that the decline in CDI rates at our hospital as well as nationwide was due to multiple factors. Further research is needed to clarify whether COVID-19 infection control measures, antibiotic usage patterns, or characteristics of hospitalized patients during the pandemic played the most important role in the decreased CDI rates observed.